# Enhancing Pre-trained ViTs for Downstream Task Adaptation: A Locality-Aware Prompt Learning Method

### Shaokun Wang
Xi'an Jiaotong University
Xi'an, China
shaokunwang.xjtu@gmail.com

### Yifan Yu
Xi'an Jiaotong University
Xi'an, China
yyf1999@stu.xjtu.edu.cn

### Yuhang He
Xi'an Jiaotong University
Xi'an, China
hyh1379478@stu.xjtu.edu.cn

### Yihong Gong*
Xi'an Jiaotong University
Xi'an, China
ygong@mail.xjtu.edu.cn

## Abstract

Vision Transformers (ViTs) excel in extracting global information from image patches. However, their inherent limitation lies in effectively extracting information within local regions, hindering their applicability and performance. Particularly, fully supervised pre-trained ViTs, such as Vanilla ViT and CLIP, face the challenge of *locality vanishing* when adapting to downstream tasks. To address this, we introduce a novel LOcality-aware pRompt lEarning (LORE) method, aiming to improve the adaptation of pre-trained ViTs to downstream tasks. LORE integrates a data-driven Black Box module (*i.e.,*a pre-trained ViT encoder) with a knowledge-driven White Box module. The White Box module is a locality-aware prompt learning mechanism to compensate for ViTs' deficiency in incorporating local information. More specifically, it begins with the design of a Locality Interaction Network (LIN), which treats an image as a neighbor graph and employs graph convolution operations to enhance local relationships among image patches. Subsequently, a Knowledge-Locality Attention (KLA) mechanism is proposed to capture critical local regions from images, learning Knowledge-Locality (K-L) prototypes utilizing relevant semantic knowledge. Afterwards, K-L prototypes guide the training of a Prompt Generator (PG) to generate locality-aware prompts for images. The locality-aware prompts, aggregating crucial local information, serve as additional input for our Black Box module. Combining pre-trained ViTs with our locality-aware prompt learning mechanism, our Black-White Box model enables the capture of both global and local information, facilitating effective downstream task adaptation. Experimental evaluations across four downstream tasks demonstrate the effectiveness and superiority of our LORE.

*Yihong Gong is the corresponding author.

## CCS Concepts

• **Computing methodologies → Transfer learning**.

## Keywords

Black-White Box model; locality-aware; knowledge-locality attention; visual prompt learning.

**ACM Reference Format:**
Shaokun Wang, Yifan Yu, Yuhang He, and Yihong Gong. 2024. Enhancing Pre-trained ViTs for Downstream Task Adaptation: A Locality-Aware Prompt Learning Method. In *Proceedings of the 32nd ACM International Conference on Multimedia (MM '24), October 28–November 1, 2024, Melbourne, VIC, Australia.* ACM, New York, NY, USA, 11 pages. https://doi.org/10.1145/3664647.3680983

## 1 Introduction

In recent years, Vision Transformers [7, 39] (ViTs) have achieved significant progress and become the mainstream of computer vision. It demonstrates strong performance on various vision tasks, including image classification [14], image retrieval [43], instance segmentation [6], *etc.* Despite its promising progress, researchers have recently identified a fundamental limitation of the ViT models, *i.e.*, though excelling at extracting global information of the image patches, ViTs are inferior in extracting information within local regions [21, 35]. This deficiency in incorporating local information can be attributed to the inherent design of the architecture, which prioritizes holistic patterns and structures while neglecting the intricate details prevalent in foreground local regions. Specifically, for a fully-supervised pre-trained ViT[1] model such as Vanilla ViT [38] and CLIP [34], it suffers from a *locality vanishing* problem when adapting to downstream tasks. As shown in Fig. 1 (a), when employing a pre-trained ViT (*e.g.*, CLIP) to extract features for several downstream task images, the majority of high-attention-weight tokens 1) are sparsely distributed in the background rather than in the foreground object region, and 2) exhibit similar and low-information semantics. This reveals that when pre-trained ViTs encounter unknown downstream task images, they primarily pay attention to global information while neglecting local information in crucial regions. Consequently, the applicability and performance of the pre-trained ViTs may be compromised when adapting a pre-trained model to downstream tasks.

---

[1]Throughout the remainder of this work, the term "pre-trained ViT" refers specifically to fully supervised pre-trained ViT.

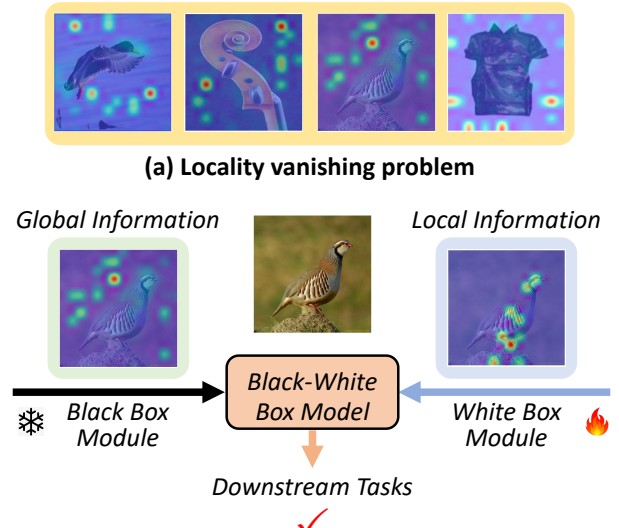

**(a) Locality vanishing problem**

*Global Information*     *Local Information*

Black Box Module ❄️    Black-White Box Model    White Box Module 🔥

Downstream Tasks ✓

**(b) Illustration of the working logic of our method**

**Figure 1: (a) In downstream tasks, pre-trained ViT tends to focus on global information while neglecting local information in critical regions, a phenomenon referred to as the locality vanishing problem. (b) Our White Box module compensates for the Black Box module's (*i.e.,*pre-trained ViT) local information incorporating capacity to better adapt to downstream tasks.**

There are methods [8, 27, 44, 48] proposed to improve the ViTs by extracting local information of the foreground objects. However, they are train-from-scratch designed, thus requiring substantial computational resources and time for deployment in downstream tasks. To overcome this deficiency, an effective strategy involves adopting Parameter-Efficient-Tuning (PET) techniques to adapt pre-trained ViTs to downstream tasks. The PET methods, such as visual prompt learning (VPL) [14, 16, 52, 53] and adapter tuning [10, 49], train only a small number of additional parameters for downstream tasks, while keeping all pre-trained ViT parameters frozen. Efficient as they are, these methods are data-driven and regard the pre-trained model as a Black Box model [40]. The inherent opacity and agnosticism of these methods present a significant challenge in comprehending the internal workings and interpreting the additional parameters of PET.

Different from the data-driven Black Box model mentioned above, White Box models [11, 17] are knowledge-driven, characterized by explicit rules and logic, facilitating interpretability of their internal workings and logical rules, while performing inferior in complex scenarios. Recent advances in a Black-White Box model theory [28, 37] suggest that combining Black Box and White Box models to construct a unified Black-White Box model provides a promising approach to enhance both the performance and interpretability of Black Box models.

Motivated by the Black-White Box model theory, we introduce a novel LOcality-aware pRompt lEarning method (LORE) aimed at enhancing the adaptation of pre-trained ViTs to downstream tasks. Our LORE consists of i) a data-driven Black Box module: *i.e.,*a pre-trained ViT encoder, and ii) a knowledge-driven White Box

module: A locality-aware prompt learning mechanism designed to compensate for pre-trained ViTs' local information incorporating capacity. More specifically, the White Box module starts with a Locality Interaction Network (LIN). The LIN treats an image as a neighbor graph, with neighbor relations among image patches represented as graph edges and image patches acting as graph nodes. Employing graph convolution, LIN enhances local relationships among image patches. Subsequently, a Knowledge-Locality Attention (KLA) is proposed to capture critical local regions from images. It utilizes relevant semantic knowledge as queries to match crucial local regions within locality-enhanced image patches, yielding Knowledge-Locality (K-L) prototypes of images. Afterwards, using a K-L prototype-guided constraint, a lightweight Prompt Generator (PG) is presented to generate *locality-aware prompts* for images. Finally, locality-aware prompts, enriched with critical local information, serve as additional input tokens for our Black Box module for local information compensation. As shown in Fig. 1 (b), combining the pre-trained ViT and locality-aware prompt learning mechanism, our Black-White Box model enables the capture of both global and local information, facilitating effective adaptation to downstream tasks.

To demonstrate the effectiveness and superiority of our proposed LORE method, we conduct comprehensive experiments on a total number of 16 benchmark datasets on 4 different downstream tasks, including image classification, image retrieval, point correspondence, and video object segmentation. On the task of image classification, the LORE steadily and significantly outperforms the existing state-of-the-art (SOTA) methods on 12 benchmarks. Besides, experiments on the other three tasks demonstrate the generality of our LORE. Ablation studies further demonstrate the effectiveness of the proposed components. In summary, the main contributions include:

- We propose a novel LOcality-aware pRompt lEarning method (LORE) consisting of a data-driven Black Box module and a knowledge-driven White Box module for downstream task adaptation.
- To mitigate the problem of locality vanishing in pre-trained ViT models, we design a locality-aware prompt learning mechanism as our White Box module to compensate for the limited local information incorporating capacity of pre-trained ViTs.
- We develop a Knowledge-Locality Attention (KLA) mechanism to capture critical local regions from images. KLA learns K-L prototypes of images utilizing a semantic knowledge-locality matching strategy, which are then leveraged to optimize the training of our Prompt Generator (PG).
- Experimental results on 4 kinds of downstream tasks, including 16 benchmark datasets, demonstrate the superiority of the proposed LORE method.

## 2 Related Work

### 2.1 VPL for Downstream Task Adaptation

Motivated by the success of prompt learning for pre-trained language models in the NLP field [18, 24, 51], investigating prompt learning for pre-trained vision models has emerged as a prominent research area. Visual prompt learning (VPL) aims to fine-tune only

a small number of task-specific parameters while freezing the entire pre-trained model [14, 23]. In comparison to alternative fine-tuning strategies (*e.g.,*Full Fine-tuning, Adapter Tuning [10, 36, 49], and Prefix Tuning [20]), VPL achieves remarkable performance and substantially reduces per-task storage requirements. Today, VPL is mainly used to adapt vision-only models [14, 42, 50] and Vision-Language Models (VLMs) [2, 15, 16, 29, 45, 52–54] to downstream tasks. Specifically, VPT [14] is the first to adopt prompt learning for pre-trained vision models. It investigates the applicability and viability of VPL and opens up an innovative avenue for downstream task adaptation. CoOp [53] proves that learnable prompts (*i.e.,*continuous prompts) perform better than hand-crafted prompts (*i.e.,*discrete prompts, like "a photo of an apple.") in terms of performance and robustness of downstream task adaptation. Maple [16] proposes multi-modal prompts to improve alignment between vision and language representations in VLMs. ProGrad [54] presents a prompt-aligned gradient method for downstream task adaptation. These methods design learnable prompts instead of hand-craft prompts and show the superiority of learnable prompts. However, it is difficult to interpret what the learnable prompts mean and how they help pre-trained models adapt to downstream tasks. In this paper, we attempt to provide a more interpretable perspective on VPL. Our locality-aware prompt learning mechanism is designed to generate locality-aware prompts, aggregating crucial local information and thus addressing the deficiency in pre-trained ViTs' capacity to incorporate local information for downstream tasks.

## 2.2 ViTs with Locality Mechanism

ViTs rely on self-attention mechanisms to extract global information among image patches [7, 21, 35, 39]. However, the lack of a locality mechanism (*e.g.,*the convolutions in CNNs) makes it difficult to capture critical local regions in foreground objects, especially in downstream tasks. The aforementioned issue restricts pre-trained ViTs' ability to adapt to downstream tasks [25]. To improve the capacity to extract local information, recent studies [9, 13, 26, 27, 31, 48] have concentrated on tokenization techniques and self-attention mechanisms. Slide-Transformer [31] proposes the slide attention for local relationship modeling. T2T-ViT [48] proposes a progressive tokenization strategy that can better encode the critical local structure for image patches. Swin Transformer [27] uses a shifted windowing scheme to provide better cross-window connections within local windows. Additionally, there has been a trend of designing hybrid architectures [8, 22, 44, 46, 47] of convolutional layers and self-attention layers in a way that local mechanisms are introduced to ViTs. For instance, ConViT [8] and CvT [44] bring locality to ViTs by adding convolutions within the transformer blocks. However, these methods are train-from-scratch designed. In light of this limitation, our LORE is intended to implement a locality-aware prompt learning mechanism for pre-trained ViTs.

## 3 Methodology

As shown in Fig. 2, our LORE consists of a Black Box module and a White Box module. Specifically, the pre-trained ViT encoder $F$ serves as the Black Box module. The locality-aware prompt learning mechanism serves as the White Box module. Within the White Box module, the Locality Interaction Network (LIN) learns locality-enhanced tokens $\hat{\mathbf{E}}$ that enhance information interaction within image local regions. Subsequently, the Knowledge-Locality Attention (KLA) is designed to capture critical local regions from $\hat{\mathbf{E}}$ under the guidance of semantic knowledge, yielding Knowledge-Locality (K-L) prototypes denoted as $\hat{\mathbf{A}}$. Utilizing LIN and K-L prototype-guided constraint $\mathcal{L}_{kp}$, the Prompt Generator (PG) is proposed to generate locality-aware prompts $\mathbf{U}$. Finally, the locality-aware prompts $\mathbf{U}$ are fed into our Black Box module alongside image patch tokens $\mathbf{E}_0$, thereby constituting a Black-White Box model. The details of the Black Box module and White Box module are presented in the following sections.

## 3.1 Black Box Module: Pre-trained ViT

We formulate our Black Box module as follows. The $F$ is a pre-trained ViT encoder with $L$ transformer layers. Given an input of image $\mathbf{X}$, the image is reshaped to $M$ flattened 2D patches. These patches are then projected into image patch tokens $\mathbf{E}_0 \in \mathbb{R}^{M \times d_e}$, where $M$ is the token length and $d_e$ is the dimension of each patch token. Furthermore, patch tokens $\mathbf{E}_0$ and a CLS token $\mathbf{C}_0$ are fed into the $F$. Formally, for the $i$-th transformer block $F_i$:

$$[\mathbf{C}_i, \mathbf{E}_i] = F_i([\mathbf{C}_{i-1}, \mathbf{E}_{i-1}]), \tag{1}$$

where $i = 1, 2, ..., L$. Notably, all parameters of the Black Box module are frozen in our method.

## 3.2 White Box Module: Locality-Aware Prompt Learning Mechanism

*3.2.1 Locality Interaction Network.* LIN first represents an image as a graph and then explicitly enhances the local relationships between neighbor image patches. Taking the $\mathbf{E}_0$ as input, we construct a directed neighbor graph $\mathcal{G} = (\mathcal{V}, \mathcal{E})$ for each image, where $\mathcal{V}$ is the node set and $\mathcal{E}$ is the edge set. More specifically, $M$ patch tokens of an image $\mathbf{E}_0 = [\mathbf{e}_1, \mathbf{e}_2, ..., \mathbf{e}_M]$ are defined as the node set $\mathcal{V}$. Each node indicates an image patch token $\mathbf{e}_i \in \mathbb{R}^{1 \times d_e}$. For a node $\mathbf{e}_i$, we establish a directed edge originating from node $\mathbf{e}_j$ to $\mathbf{e}_i$ when $\mathbf{e}_j$ is identified as one of the TopK nearest neighbors of $\mathbf{e}_i$, denoted as $\mathbf{e}_j \in \mathcal{N}(\mathbf{e}_i)$. Obviously, this graph $\mathcal{G}$ represents local relationships between image patch tokens, which can be used as a prior to help characterize local regions. Afterwards, we use the max-relative graph convolution $\Theta$ [12, 19] to enhance the locality interaction between an image patch token $\mathbf{e}_i$ and its neighbors $\mathcal{N}(\mathbf{e}_i)$. In this way, the $\mathbf{E}_0$ can be updated to $\mathbf{E}' \in \mathbb{R}^{M \times d_e}$:

$$\begin{aligned} \mathbf{E}' &= \Theta(\mathbf{E}_0, \mathcal{G}) \\ &= [\mathbf{e}'_1, \mathbf{e}'_2, ..., \mathbf{e}'_i, ..., \mathbf{e}'_M], \end{aligned} \tag{2}$$

with

$$\mathbf{e}'_i = max(\mathbf{e}_i - \mathbf{e}_j | \mathbf{e}_j \in \mathcal{N}(\mathbf{e}_i)), \tag{3}$$

where $max(\cdot)$ is a max-pooling feature aggregator to pool the difference of features between $\mathbf{e}_i$ and its neighbors. Eq. (3) further details the operation of $\Theta$ from the perspective of an image patch token $\mathbf{e}_i$. And then, we design a residual connect module $h(\cdot, \cdot)$ to alleviate over-smoothing problem [5] of the max-relative graph convolution. Finally, the locality-enhanced tokens $\hat{\mathbf{E}} \in \mathbb{R}^{M \times d_e}$ of an image is defined as:

$$\hat{\mathbf{E}} = h(\mathbf{E}', \mathbf{E}_0) = \sigma(\mathbf{E}'\mathbf{W}_1)\mathbf{W}_2 + \mathbf{E}_0, \tag{4}$$

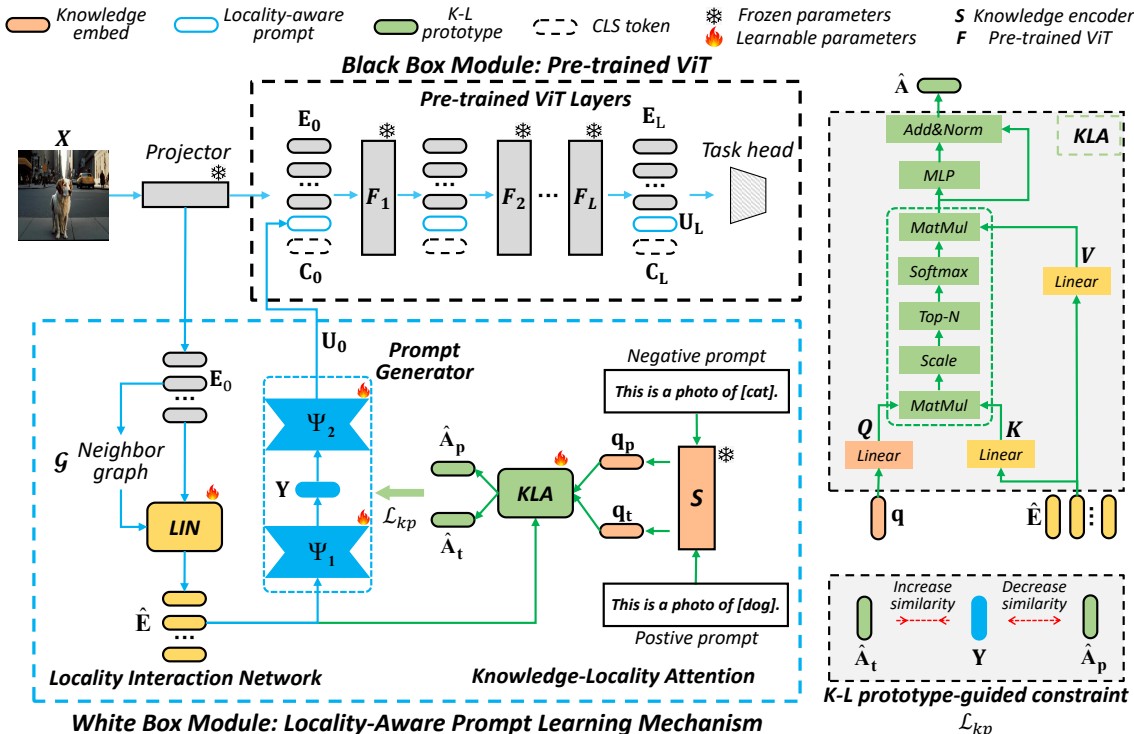

**Figure 2: The framework of our LORE. We design the locality-aware prompt learning mechanism (*i.e.,*White Box module) to compensate for the local information incorporating capacity of pre-trained ViTs (*i.e.,*Black Box module). Our White Box module consists of a Locality Interaction Network (LIN), a Knowledge-Locality Attention (KLA), and a Prompt Generator (PG). The workflow indicated by the green lines is not necessary for the inference phase.**

where $\sigma$ is the relu non-linearity, $\mathbf{W}_1 \in \mathbb{R}^{d_e \times d_e}$ and $\mathbf{W}_2 \in \mathbb{R}^{d_e \times d_e}$ are fully-connected layers. $\hat{\mathbf{E}}$ enhances the local relationships between image patches in comparison to $\mathbf{E}_0$.

*3.2.2 Knowledge-Locality Attention.* $\hat{\mathbf{E}}$ includes the entire regions of an image. There are still a few local regions in $\hat{\mathbf{E}}$ that are unimportant in terms of representing the image's salient characteristics. To this end, we introduce the KLA, guided by relevant semantic knowledge, which helps in capturing important local regions of $\hat{\mathbf{E}}$ while discarding unimportant ones.

We formulate the problem of capturing important local regions as a semantic knowledge-locality matching between the semantic knowledge embedding $\mathbf{q}$ and the locality-enhanced tokens $\hat{\mathbf{E}}$. More specifically, we encode auxiliary prompts into semantic embeddings $\mathbf{q}$ utilizing a semantic knowledge encoder denoted as $S$, represented as $\mathbf{q} = S(\mathbf{Prompt})$. These auxiliary prompts contain relevant semantic knowledge corresponding to the foreground objects in images, which are handcrafted instructions like "This is a photo of a [CLASS].". In KLA, we use $\mathbf{q}$ as the input of the Q (query) and $\hat{\mathbf{E}}$ as the input of the K (key) and V (value). Then we apply linear transformations to generate Q, K, and V, respectively:

$$\mathbf{Q} = \mathbf{q}\mathbf{W}_{\mathbf{q}}, \mathbf{K} = \hat{\mathbf{E}}\mathbf{W}_{\mathbf{k}}, \mathbf{V} = \hat{\mathbf{E}}\mathbf{W}_{\mathbf{v}}, \tag{5}$$

where $\mathbf{W}_{\mathbf{q}} \in \mathbb{R}^{d_q \times d_e}$, $\mathbf{W}_{\mathbf{k}} \in \mathbb{R}^{d_e \times d_e}$, $\mathbf{W}_{\mathbf{v}} \in \mathbb{R}^{d_e \times d_e}$, $\mathbf{Q} \in \mathbb{R}^{1 \times d_e}$, $\mathbf{K} \in \mathbb{R}^{M \times d_e}$, and $\mathbf{V} \in \mathbb{R}^{M \times d_e}$. Furthermore, we define an $ATTN(\cdot)$ function to match and aggregate the crucial local regions into $\mathbf{A}$:

$$\mathbf{A} = ATTN(\mathbf{Q}, \mathbf{K}, \mathbf{V}) = Softmax(\varphi_N(\mathbf{Q}\mathbf{K}^{\top}/\sqrt{d_e}))\mathbf{V}, \tag{6}$$

where $\sqrt{d_e}$ is a scaling factor. The $\varphi_N(\cdot)$ denotes a row-wise top-N filter, which sets the top-N values unchanged and the rest to 0, aiming to ensure that $ATTN(\cdot)$ focuses on the most relevant locality-enhanced tokens. For simplicity, the concept of multiple heads here is omitted. In Eq.(6), matrix product of $\mathbf{Q}\mathbf{K}^{\top} \in \mathbb{R}^{1 \times M}$ represents the similarity between the semantic embedding $\mathbf{q}$ and all locality-enhanced tokens $\hat{\mathbf{E}}$ of an image. Finally, a multi-layer perceptron (*MLP*) and a residual connection are adopted to enhance the representation ability of $\mathbf{A}$. The Knowledge-Locality (K-L) prototype $\hat{\mathbf{A}}$ is formulated as:

$$\hat{\mathbf{A}} = BN(MLP(\mathbf{A}) + \mathbf{A}), \tag{7}$$

where $BN$ indicates the batch normalization operation. This K-L prototype $\hat{\mathbf{A}}$ contains critical local information about an image.

*3.2.3 Prompt Generator.* Taking the K-L prototype $\hat{\mathbf{A}}$ as input, a straightforward method for generating locality-aware prompt $\mathbf{U}$ is to directly feed $\hat{\mathbf{A}}$ into the PG. However, this solution necessitates the model to compute the corresponding K-L prototype for each image during inference, increasing computation complexity and requiring auxiliary prompts for the prototype generation, which is impractical for real-world applications. To overcome this limitation, we develop a K-L prototype-guided constraint $\mathcal{L}_{kp}$. It facilitates PG training with the use of the K-L prototypes $\hat{\mathbf{A}}$, wherein K-L prototypes are not necessary for the inference phase. We firstly formulate the architecture of the PG and then describe the details of $\mathcal{L}_{kp}$.

PG has two lightweight bottleneck architectures. These bottleneck architectures reduce the number of parameters while ensuring effectiveness. Using the first bottleneck architecture $\Psi_1$ parameterized by $\theta_1$, we map the $\hat{E}$ to a latent space. We define a latent embedding $Y \in \mathbb{R}^{1 \times d_e}$ as:

$$Y = \Psi_1(avp(\hat{E}); \theta_1), \qquad (8)$$

where $avp(\cdot)$ is the average pooling operation and $\Psi_1$ consists of two $1 \times 1$ convolutions. Initially, it reduces the feature dimension to $r$, followed by an expansion of the dimension back to $d_e$. $r \ll d_e$. Afterwards, the second bottleneck architecture $\Psi_2$ parameterized by $\theta_2$ is used to generate the locality-aware prompt $U$:

$$U = \Psi_2(Y; \theta_2). \qquad (9)$$

$\Psi_2$ comprises two $1 \times 1$ convolution operations that initially decrease the feature dimension to $r$ and then expand the dimension back to $d_e$. The normalization operations are omitted here.

K-L prototype-guided constraint $\mathcal{L}_{kp}$ optimizes the training objective of the PG. More specifically, we first introduce the positive and negative K-L prototypes. For a downstream task image $X$ with a true label $t$, we can obtain a predicted label $p$ using our Black Box module. $p = t$ denotes a correct prediction, whereas $p \neq t$ denotes a wrong prediction. When $p \neq t$, we define the positive / negative prompt like "This is a photo of a [CLASS $t$ / CLASS $p$].". For instance, for a "dog" image of the downstream task, the B-B module incorrectly predicts it as a "cat". We verbalize the wrong prediction as a negative prompt, such as "This is a photo of a cat", which demonstrates the B-B's wrong understanding of this image. We hope to correct the wrong understanding during PG's training phase. To this end, we compute the corresponding positive/negative K-L prototype $\hat{A}_t / \hat{A}_p$ using Eq. (5) and (6). $\hat{A}_p$ is the negative K-L prototype that contains misleading local information about the image. And then, we define $\mathcal{L}_{kp}$ to guide the latent embedding $Y$ that are similar to $\hat{A}_t$ but not similar to $\hat{A}_p$ in a joint latent space:

$$\mathcal{L}_{kp} = -cos(Y, \hat{A}_t) + \mathbb{I}(t \neq p) * cos(Y, \hat{A}_p), \qquad (10)$$

where $\mathbb{I}(\cdot)$ is the indicator function. If $t \neq p$ is true, $\mathbb{I}(t \neq p)$ is 1. Otherwise, $\mathbb{I}(t \neq p)$ is 0. $cos(\cdot, \cdot)$ is the cosine similarity.

### 3.3  Training and Inference Pipeline of LORE

During the training phase, the overall training loss of our model is formulated as:

$$\mathcal{L} = \mathcal{L}_{task} + \lambda \mathcal{L}_{kp}, \qquad (11)$$

where $\mathcal{L}_{task}$ is the downstream task loss. $\mathcal{L}_{kp}$ is used to optimize the training of PG, and we describe the details of $\mathcal{L}_{kp}$ in Eq.(10). $\lambda$ indicates the hyper-parameter to balance the contributions of the two losses. The LIN, PG, and KLA are trainable.

During the inference phase, we first compute the locality-enhanced tokens $\hat{E}$ for an image by using Eq. (2), (3), and (4). And then, Eq. (8) and (9) are used to generate the locality-aware prompt $U$. Finally, $U$ is used as additional input tokens containing the critical local information, fed into our pre-trained ViT along with image patch tokens. The $i$-th transformer block $F_i$ in our model is formulated as:

$$[C_i, U_i, E_i] = F_i([C_{i-1}, U_{i-1}, E_{i-1}]), \qquad (12)$$

where $i = 1, 2, ..., L$. The outputs of $F_L$ are utilized to get downstream task results.

## 4  Experiments

In the following part of this section, we first provided the experimental setup, then presented the evaluation results and visualization results, and finally showed ablation studies.

### 4.1  Experimental Setup

**Datasets.** For the classification task, we conducted experiments on three kinds of datasets: (1) Natural datasets: CIFAR-10, CIFAR-100, DTD, and ImageNet. (2) Fine-grained datasets: Flowers102, Stanford-Cars, FGVCAircraft, and StanfordDogs. (3) Specialized datasets: EuroSAT, Resisc45, UCF101, and Pattern. For the image retrieval task, we performed experiments on ROxford5k and RParis6k datasets. For the point correspondences task, we reported results on SPair-71k dataset. For the video object segmentation task, we conducted experiments on DAVIS 2017 dataset.

**Implementation Details.** We implemented our LORE[2] in Py-Torch with two NVIDIA RTX 3090 GPUs. We adopted the visual encoder of pre-trained CLIP ViT-B/16 [34], pre-trained CLIP ViT-B/32 [34], and Vanilla ViT-B/16 pre-trained on ImageNet-21k [38] as our Black Box modules, respectively. We adopted the text encoder of pre-trained CLIP as our semantic knowledge encoder. Our models are trained using the SGD optimizer with a mini-batch size of 32/64. In the few-shot classification task and the first stage of the easy-to-hard classification task, we set the initial learning rate to 0.003 and decreased it to 0.0001 by the cosine annealing rule. In the second stage of the easy-to-hard classification task, we set the initial learning rate to 0.0005 and decrease it to 0.0001 by the cosine annealing rule. We trained our method for 50 epochs. We followed the same data augmentation strategies in our comparison methods. The hyper-parameter $\lambda$ is set to 0.1. The length of locality-aware prompts is set to 10. The dimension $r$ in $\Psi_1$ and $\Psi_2$ is set to 32.

### 4.2  Results of Downstream Task Adaptation

*4.2.1  Classification.* We compared our model with SOTA prompt learning methods ( CoOp [53], Co-CoOp [52], Maple [16], and Pro-Grad [54]), representative adapter-based method ( Clip-Adapter [10]), and Zero-shot CLIP [34]. For a fair comparison, we adopted the same pre-trained CLIP as the base model of our comparison methods, and we used the dataset split strategy as the same as our method to reproduce these methods. In this task, LORE's task head is trainable.

**Few-shot classification.** Fine-tuning models with few-shot training samples that are randomly selected, followed by performance evaluation on the total test set. Notably, we used the same training samples in our comparison methods.

We compared our LORE with all these methods on 12 datasets with 16-shot and 8-shot settings. As shown in Table 1, our LORE outperforms the SOTA methods with the 16-shot setting: (1) it has achieved the highest accuracies on all datasets. (2) it has achieved the highest average accuracy. LORE surpasses the best method Clip-Adapter by 7.09% in average accuracy. Besides, our LORE also outperforms these methods with the 8-shot setting: (1) it has

---

[2]The code is available at https://github.com/Mysteriousplayer/KGPT.

**Table 1: Comparison with the SOTA methods on 12 datasets under the few-shot classification task, where the Black Box module is CLIP ViT-B/16. Our LORE has achieved the highest average accuracy.**

| Method | Flowers102 | StanfordCars | Aircraft | StanfordDogs | CIFAR-10 | CIFAR-100 | DTD | ImageNet | EuroSAT | Resisc45 | UCF101 | Pattern | Average | Shot |
|---|---|---|---|---|---|---|---|---|---|---|---|---|---|---|
| CLIP | 71.30 | 63.84 | 24.72 | 62.67 | 88.38 | 64.78 | 43.40 | 66.59 | 35.80 | 62.60 | 66.72 | 61.33 | 59.34 | 0 |
| Co-CoOp | 90.42 | 73.72 | 35.70 | 71.15 | 80.00 | 55.19 | 65.53 | 71.02 | 73.02 | 81.81 | 79.46 | 91.23 | 72.35 | |
| ProGrad | 95.78 | 74.87 | 38.34 | 70.33 | 80.66 | 54.97 | 68.14 | 71.13 | 76.16 | 83.11 | 77.13 | 93.52 | 73.68 | |
| Clip-Adapter | 96.59 | 77.99 | 43.41 | 69.64 | 80.86 | 55.33 | 69.73 | 70.02 | 78.96 | 86.44 | 82.77 | 95.71 | 75.62 | 16-shot |
| CoOp | 96.79 | 79.84 | 43.05 | 72.65 | 78.80 | 54.57 | 68.09 | 71.51 | 78.68 | 84.70 | 82.37 | 94.85 | 75.49 | |
| Maple | 93.59 | 76.58 | 39.48 | 75.08 | 80.72 | 62.84 | 69.47 | 70.72 | 79.57 | 83.26 | 81.89 | 93.82 | 75.59 | |
| LORE | **98.98** | **87.20** | **54.34** | **78.14** | **93.19** | **72.39** | **75.00** | **72.16** | **87.04** | **88.94** | **87.13** | **98.04** | **82.71** | |
| Co-CoOp | 88.02 | 71.51 | 32.46 | 70.09 | 78.50 | 55.03 | 59.15 | 70.19 | 65.20 | 78.06 | 76.79 | 86.91 | 69.33 | |
| ProGrad | 93.54 | 73.37 | 34.53 | 69.78 | 79.08 | 53.50 | 62.93 | 70.44 | 69.74 | 80.29 | 76.08 | 90.84 | 71.18 | |
| Clip-Adapter | 94.03 | 73.64 | 36.57 | 66.60 | 79.35 | 53.95 | 64.15 | 69.02 | 71.99 | 82.29 | 80.70 | 92.50 | 72.07 | 8-shot |
| CoOp | 95.01 | 76.50 | 36.57 | 70.01 | 78.67 | 52.97 | 64.52 | 69.19 | 70.16 | 81.38 | 79.88 | 91.45 | 72.19 | |
| Maple | 90.05 | 72.62 | 32.91 | **72.45** | 79.99 | 61.00 | 64.89 | 70.06 | 66.98 | 79.07 | 79.12 | 86.97 | 71.34 | |
| LORE | **97.48** | **81.25** | **42.99** | 70.72 | **87.79** | **64.75** | **70.21** | **70.65** | **76.95** | **84.25** | **83.48** | **96.51** | **77.25** | |

achieved the highest accuracies or compatible ones on all datasets. (2) it has achieved the highest average accuracy. Compared to the best method CoOp, our LORE surpasses it by 5.06% in average accuracy. Moreover, as we can see from Table 2, LORE also has achieved the highest average accuracy when we use pre-trained ViT-B/32 as the Black Box module: (1) LORE surpasses the best methods by 6.82%, 5.52%, and 3.69% on 4 fine-grained datasets, 4 natural datasets, and 4 specialized datasets with the 16-shot setting, respectively. (2) LORE outperforms the best methods with a gap of 4.67%, 2.04%, and 3.43% on 3 kinds of datasets with the 8-shot setting, respectively.

**Easy-to-Hard classification.** To test whether our White Box module can improve pre-trained ViTs when encountered with hard samples, we devise a two-stage classification task comprising an easy curriculum stage and a hard curriculum stage. More specifically, we assess sample difficulty based on their distribution in the

**Table 2: Comparison with the SOTA methods on 3 kinds of datasets under the few-shot classification task, where the Black Box module is CLIP ViT-B/32.**

| Method | Fine-grained (4) | Natural (4) | Specialized (4) | Average | Shot |
|---|---|---|---|---|---|
| CLIP | 49.96 | 63.86 | 51.90 | 55.24 | 0 |
| Co-CoOp | 59.41 | 63.20 | 78.65 | 67.09 | |
| ProGrad | 63.91 | 64.48 | 79.52 | 69.30 | |
| Clip-Adapter | 65.85 | 63.77 | 83.85 | 71.16 | 16-shot |
| CoOp | 65.95 | 64.68 | 82.86 | 71.16 | |
| Maple | 64.49 | 68.37 | 83.32 | 72.06 | |
| LORE | **72.77** | **73.89** | **87.54** | **78.06** | |
| Co-CoOp | 55.91 | 62.08 | 74.23 | 64.08 | |
| ProGrad | 60.97 | 62.59 | 74.02 | 65.86 | |
| Clip-Adapter | 60.91 | 62.72 | 79.05 | 67.56 | 8-shot |
| CoOp | 61.37 | 62.17 | 78.19 | 67.24 | |
| Maple | 60.01 | 65.47 | 77.30 | 67.59 | |
| LORE | **66.04** | **67.51** | **82.48** | **72.01** | |

feature space of our Black Box module. For a sample $\mathbf{X}$ belonging to class $j$, the difficulty function $\mathbf{D}$ is definable using the distance between its feature (*i.e.,*CLS token) $\mathbf{C}_L$ and its class centroid $\mathbf{o}_j$: $\mathbf{D} = dis(\mathbf{C}_L, \mathbf{o}_j)$, where $dis(\cdot, \cdot)$ is the cosine distance. A lower difficulty score indicates that $\mathbf{X}$ is closer to its class centroid $\mathbf{o}_j$ and is thus considered an easy sample, suitable for initial adaptation to downstream tasks. Conversely, a higher difficulty score suggests that $\mathbf{X}$ is farther from its class centroid $\mathbf{o}_j$ and is more likely to be misclassified, making it a suitable candidate for further adaptation of the model to downstream tasks. In the easy curriculum stage, we select the $N$ easiest samples to train the LORE-e. Subsequently, in the hard curriculum stage, we select the $N$ hardest samples based on the LORE-e. These selected samples are then utilized to further train the LORE-h. Importantly, both LORE-e and LORE-h share an identical framework. It is worth noting that the training samples of each class are given 16/stage and 8/stage, respectively. We did not test the 16/stage setting on Flowers102, StanfordCars, DTD, and UCF101. Because the number of training samples in the 4 datasets is not insufficient to select 16 qualified hard samples.

According to Table 3, our method has shown obvious advantages over the SOTA methods: (1) our LORE-h has obtained the highest accuracies on all datasets and the highest average accuracies for the 16/stage and 8/stage settings; (2) our LORE-e has achieved the highest average accuracies for the 16/stage and 8/stage settings; (3) our LORE-e has achieved the highest accuracies or compatible one on all datasets for the two settings; (4) our LORE-h outperforms our LORE-e by 1.66% and 2.04% for both settings, respectively, which is the most significant improvement compared to SOTA methods. In conclusion, our LORE-h outperforms our LORE-e, demonstrating the superiority of our locality-aware prompt learning mechanism in improving the performance of our Black Box module when encountered with difficult samples.

*4.2.2 Effectiveness on Other Downstream Tasks.* To further verify the effectiveness of our Black-White Box model in improving pre-trained ViTs, we conduct experiments on other downstream tasks. Our White Box module, pre-trained on the classification task,

**Table 3: Comparison with the SOTA methods on 12 datasets under the easy-to-hard classification task, where the Black Box module is CLIP ViT-B/16. 'xxx-e' and 'xxx-h' represent being trained only in the easy curriculum stage and being trained in the easy curriculum stage followed by the hard curriculum stage, respectively. The highest accuracy in the easy/hard curriculum is indicated with an underline / in bold. Absolute improvements from the easy to hard curriculum are indicated in parentheses.**

| Method | Dataset | | | | | | | | | | | | | Shot |
| | Flowers | Cars | Aircraft | Dogs | CF-10 | CF-100 | DTD | ImageNet | SAT | Resisc | UCF | Pattern | Avg. | |
| --- | --- | --- | --- | --- | --- | --- | --- | --- | --- | --- | --- | --- | --- | --- |
| CLIP | 71.30 | 63.84 | 24.72 | 62.67 | 88.38 | 64.78 | 43.40 | 66.59 | 35.80 | 62.60 | 66.72 | 61.33 | 59.34 | 0 |
| CoOp-e | - | - | 44.34 | 75.59 | 82.50 | 58.72 | - | 71.30 | 80.84 | 86.29 | - | 94.74 | 74.29 | 16/stage |
| CoOp-h | - | - | 45.33 | 75.19 | 78.80 | 58.02 | - | 70.75 | 64.69 | 85.35 | - | 89.59 | 70.97(-3.32) | |
| Co-CoOp-e | - | - | 35.43 | 71.68 | 82.54 | 56.29 | - | 71.27 | 72.32 | 82.05 | - | 90.41 | 70.25 | |
| Co-CoOp-h | - | - | 37.02 | 71.95 | 81.52 | 54.95 | - | 71.01 | 60.72 | 82.30 | - | 89.85 | 68.67(-1.58) | |
| Clip-Adapter-e | - | - | 46.56 | 71.74 | 82.26 | 56.27 | - | 70.23 | 81.27 | 87.70 | - | 95.86 | 73.99 | |
| Clip-Adapter-h | - | - | 47.37 | 71.67 | 82.23 | 56.55 | - | 70.00 | 81.26 | 87.10 | - | 96.05 | 74.03(+0.04) | |
| ProGrad-e | - | - | 39.48 | 71.00 | 80.26 | 55.99 | - | 71.68 | 79.89 | 84.20 | - | 94.26 | 72.10 | |
| ProGrad-h | - | - | 43.74 | 74.95 | 79.88 | 57.86 | - | 70.59 | 74.95 | 84.55 | - | 93.03 | 72.44(+0.34) | |
| Maple-e | - | - | 38.85 | 75.41 | 81.81 | 62.41 | - | 72.21 | 82.41 | 84.11 | - | 93.24 | 73.81 | |
| Maple-h | - | - | 41.31 | 75.52 | 80.79 | 62.36 | - | 71.24 | 85.16 | 84.31 | - | 94.08 | 74.35 (+0.54) | |
| LORE-e | - | - | 56.92 | 77.25 | 91.71 | 76.53 | - | 76.64 | 88.07 | 88.02 | - | 96.96 | 81.26 | |
| LORE-h | - | - | **61.00** | **79.80** | **92.88** | **77.56** | - | **75.73** | **89.05** | **89.67** | - | **97.66** | **82.92 (+1.66)** | |
| CoOp-e | 95.62 | 79.11 | 41.49 | 72.27 | 81.42 | 56.70 | 68.99 | 71.22 | 77.83 | 84.15 | 81.10 | 93.14 | 75.25 | 8/stage |
| CoOp-h | 95.25 | 78.88 | 41.31 | 71.56 | 81.30 | 55.34 | 62.93 | 70.48 | 72.17 | 82.64 | 79.25 | 89.05 | 73.43(-1.82) | |
| Co-CoOp-e | 86.60 | 72.30 | 32.49 | 70.96 | 80.67 | 55.30 | 62.45 | 70.87 | 68.00 | 80.27 | 78.30 | 87.71 | 70.49 | |
| Co-CoOp-h | 88.39 | 72.38 | 33.51 | 70.17 | 76.24 | 54.35 | 59.31 | 70.25 | 63.83 | 78.36 | 77.61 | 88.08 | 69.37(-1.12) | |
| Clip-Adapter-e | 95.74 | 76.53 | 42.33 | 69.51 | 80.52 | 54.28 | 69.41 | 69.37 | 76.99 | 85.79 | 82.21 | 94.03 | 74.73 | |
| Clip-Adapter-h | 95.41 | 76.63 | 43.26 | 70.75 | 81.30 | 55.36 | 69.79 | 68.93 | 74.93 | 85.51 | 82.63 | 94.36 | 74.91(+0.18) | |
| ProGrad-e | 94.80 | 72.07 | 36.48 | 69.38 | 79.88 | 54.07 | 65.11 | 71.29 | 75.11 | 82.41 | 74.20 | 90.16 | 72.08 | |
| ProGrad-h | 94.32 | 76.92 | 41.67 | 72.03 | 81.09 | 56.36 | 66.60 | 70.03 | 70.89 | 82.58 | 80.10 | 90.62 | 73.60(+1.52) | |
| Maple-e | 89.85 | 73.91 | 35.16 | 72.19 | 80.93 | 60.64 | 66.76 | 71.36 | 70.58 | 80.23 | 79.22 | 87.58 | 72.37 | |
| Maple-h | 91.47 | 75.75 | 36.12 | 72.94 | 80.37 | 60.98 | 68.03 | 70.72 | 75.37 | 81.02 | 79.75 | 90.84 | 73.61 (+1.24) | |
| LORE-e | 97.16 | 83.14 | 49.62 | 74.95 | 89.11 | 69.69 | 73.40 | 71.76 | 83.59 | 87.00 | 84.32 | 95.99 | 79.98 | |
| LORE-h | **97.77** | **85.42** | **54.94** | **78.21** | **91.43** | **72.73** | **74.36** | **73.34** | **86.32** | **87.97** | **85.14** | **96.58** | **82.02 (+2.04)** | |

demonstrates good generalization ability across different downstream tasks, including image retrieval, point correspondences, and video object segmentation.

**Image Retrieval.** We used the evaluation protocols of [3] to compare the performance of off-the-shelf features in our Black-White Box model and the Black Box model. Concretely, the pre-trained feature (*i.e.,*CLS token) is frozen and used directly for image retrieval using the K-NN strategy without any fine-tuning. We reported the Mean Average Precision (mAP) for the Medium (M) and Hard (H) split on ROxford5k and RParis6k datasets [33]. As depicted in Table 4, our Black-White Box models have clear advantages over Black Box models (*i.e.,*Vanilla ViT-B/16 and CLIP ViT-B/16).

**Point Correspondences.** In the point correspondences task, given a source image with annotated keypoints, the objective is to predict the locations of corresponding keypoints in a target image. We followed the evaluation protocols outlined in [1, 41]. More specifically, image patch tokens are mapped to the size of the original image by bi-linear interpolation. We compute the key point features in the source image and match the mutual nearest neighbors in the target image. In Table 4, we reported the Percentage of Correct Keypoint (PCK) on SPair-71k [30] dataset. Our Black-White Box models outperform their Black Box counterparts.

**Video Object Segmentation.** We conducted experiments on DAVIS 2017 [32] for the task of video object segmentation. It is a semi-supervised task that aims to propagate the first frame's segmentation mask to subsequent frames. We followed the evaluation protocols outlined in [4]. Image patch tokens of video frames are extracted to segment scenes without any fine-tuning. As illustrated in Table 4, we reported the mean region similarity and contour-based accuracy (J&F M) of our Black-White Box models and the corresponding Black Box models. We can observe that our Black-White Box models perform better in this dense recognition task.

### 4.3 Visualization

We reported visualization results to make intuitive explanations for our method. Specifically, we visualized the self-attention maps generated by our Black Box module and our Black-White Box model. These self-attention maps indicate the self-attention of the CLS token and other image tokens across the heads of the last ViT layer. Additionally, we visualized the attention map of our KLA mechanism, which illustrates the attention between the semantic knowledge embedding $\mathbf{q}$ and the locality-enhanced tokens $\hat{\mathbf{E}}$. As shown in Fig. 3, the Black Box module focuses on global information while neglecting local information in critical regions, illustrating the locality vanishing problem. Conversely, our KLA mechanism effectively directs attention towards critical local regions. Our Black-White Box model enables the extraction of both global and local information, providing a comprehensive method for capturing diverse spatial dependencies. Significantly, throughout the training phase, the parameters of the Black Box module remain

**Table 4: Evaluations under other downstream tasks, including image retrieval, point correspondence and video object segementation, where the Black Box (B-B) modules are CLIP ViT-B/16 and Vanilla ViT-B/16.**

| Task | Image Retrieval | | | | | | | | Point Correspondences | | Video Object Segmentation | |
|---|---|---|---|---|---|---|---|---|---|---|---|---|
| Dataset | ROxford5k | | | | RParis6k | | | | SPair-71k | | Davis | |
| Metric | CLIP | | ViT | | CLIP | | ViT | | CLIP | ViT | CLIP | ViT |
| | M | H | M | H | M | H | M | H | PCK@0.1 | | J&F M | |
| B-B | 0.397 | 0.107 | 0.302 | 0.094 | 0.708 | 0.482 | 0.603 | 0.358 | 18.25 | 16.61 | 54.38 | 58.12 |
| LORE | **0.418** | **0.171** | **0.449** | **0.172** | **0.750** | **0.552** | **0.720** | **0.523** | **20.71** | **18.07** | **55.62** | **59.46** |

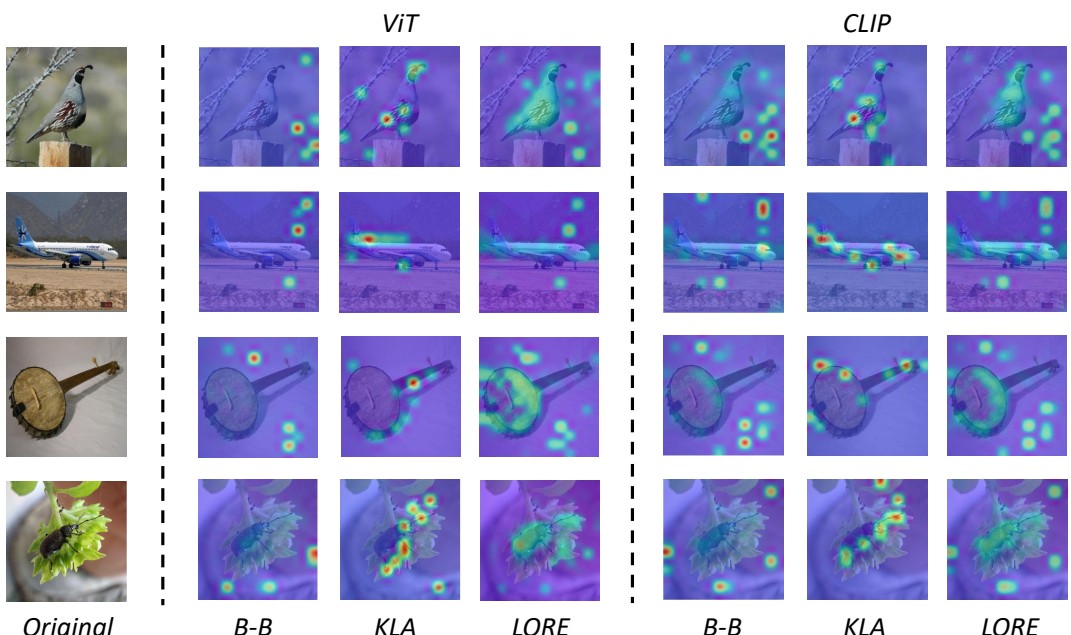

**Figure 3: Attention map visualization of our Black Box module (B-B), our KLA mechanism, and our LORE on ImageNet under the classification task. Our LORE achieves a balancing of global and local information within downstream tasks.**

**Table 5: Ablation studies of our LORE on 4 natural datasets. B-B, W-B, $\mathcal{L}_{kp}$, and VPT indicate the Black Box module, the White Box module, the K-L prototype-guided constraint, and the visual prompt tuning [14], respectively.**

| Method | W-B | $\mathcal{L}_{kp}$ | VPT | Avg. Acc. | |
|---|---|---|---|---|---|
| | | | | ViT | CLIP |
| B-B with VPT | ✗ | ✗ | ✓ | 74.01 | 73.91 |
| LORE w/o $\mathcal{L}_{kp}$ | ✓ | ✗ | ✓ | 75.24 | 77.16 |
| LORE | ✓ | ✓ | ✓ | **76.38** | **78.19** |

entirely frozen within the Black-White Box model. Consequently, global information persists within our LORE, albeit with reduced attention weights. Moreover, we think that this preservation is not deleterious; retaining global information to a certain degree can be beneficial for the downstream task adaptation.

### 4.4 Ablation Studies

We designed ablation studies to further investigate the effectiveness of our method. Ablation studies are conducted on the 16-shot classification task. The Black Box modules here are Vanilla ViT-B/16 and CLIP ViT-B/16. First, we removed the White Box module and adopted visual prompt tuning [14] to fine-tune the Black Box module (*i.e.,*"B-B with VPT"), which is a commonly used prompt

learning method. Second, we removed the K-L prototype-guided constraint $\mathcal{L}_{kp}$ to obtain another baseline "LORE w/o $\mathcal{L}_{kp}$". Notably, our KLA is also removed in "LORE w/o $\mathcal{L}_{kp}$". As shown in Table 5, when using the Vanilla ViT-B/16 as the Black Box module, our method achieves the best performance compared with "LORE w/o $\mathcal{L}_{kp}$" and "B-B + VPT", leading to improvements of 1.14% and 2.37%, respectively. When using the CLIP ViT-B/16 as the Black Box module, LORE outperforms the two baselines by 1.03% and 4.28%, respectively. These results demonstrate that the White Box module helps adapt pre-trained ViTs to downstream tasks.

### 5 Conclusions and Future Works

In this work, we propose a locality-aware prompt learning method for downstream task adaptation. Specifically, we utilize the pre-trained ViT encoder as our Black Box module, while designing the locality-aware prompt learning mechanism, referred to as the White Box module. In our White Box module, the LIN, KLA, and PG collaborate to generate the locality-aware prompts, which can enhance the local information incorporating capacity of the Black Box module. We showcase LORE's superiority and effectiveness on 4 downstream tasks. A promising future research direction is utilizing rules from external knowledge graphs to develop more interpretable White Box modules.

## Acknowledgments

This work is supported in part by the National Key Research and Development Project of China under Grant 2020AAA0105600, in part by the National Natural Science Foundation of China under Grant No. U21B2048 and No. 62302382, in part by the Shenzhen Key Technical Projects under Grant CJGJZD2022051714160501, in part by the China Postdoctoral Science Foundation under Grant No.2024M752584, and in part by the CAAI-MindSpore Open Fund, developed on OpenI Community.

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

## A   SUPPLEMENTARY MATERIAL

### A.1   Datasets statistics

The comprehensive statistics of the classification datasets are presented in Table 6. For StanfordDogs, CIFAR-10, CIFAR-100, DTD, ImageNet, Resisc45, and Pattern, we followed the official dataset split strategy. For Flowers102, StanfordCars, Aircraft, EuroSAT, and UCF, we followed the split strategy used in CoOp.

For image retrieval datasets, ROxford5k and RParis6k contain 4,993 and 6,322 high-resolution (1024 × 768) images, respectively, and each dataset has 70 queries from 11 landmarks.

For point correspondences dataset, SPair-71k comprises 70,958 image pairs from 18 classes with diverse variations in viewpoint and scale, of which 53340 pairs serve as the training set, 5384 pairs serve as the validation set, and 12234 pairs serve as the test set.

For video object segmentation dataset, DAVIS consists of 50 video sequences with 3455 densely annotated frames in pixel level. 30 videos with 2079 frames are for training, and 20 videos with 1376 frames are for validation.

### A.2   Computational complexity analysis

As shown in Table 7, the number of trainable parameters, GFLOPs, and FPS of our LORE during the inference phase remain comparable to those of "B-B" and "B-B+VPT". Moreover, our LORE significantly improves performance on downstream tasks and offers an interpretable perspective for visual prompt learning. Therefore, we believe these trade-offs are justified.

### A.3   Limitations

Our LORE aims to enhance the adaptation of pre-trained ViTs to downstream tasks. Therefore, the primary objective of our experimental design is to validate the effectiveness of pre-trained ViTs in adapting to downstream tasks. To ensure comprehensive evaluations, we compared our LORE with several effective and representative PET methods, including CoOp, Co-CoOp, Maple, ProGrad, Clip-adapter, and VPT. Some of these methods have demonstrated outstanding performance in domain generalization and cross-dataset transfer evaluations using pre-trained CLIP models. However, it is important to note that domain generalization and cross-dataset transfer evaluations assess the CLIP-based model's generalization ability, which is beyond the scope of this study. We would like to further investigate the generalization ability problem in future work.

**Table 6: Classification datasets statistics.**

| Dataset | Description | Classes | Train | Test |
|---------|-------------|---------|-------|------|
| Flowers102 | | 102 | 5726 | 2463 |
| Stanford Cars | Fine-grained | 196 | 8144 | 8041 |
| Aircraft | | 100 | 6667 | 3333 |
| Stanford Dogs | | 120 | 12000 | 8580 |
| CIFAR-10 | | 10 | 50000 | 10000 |
| CIFAR-100 | Natural | 100 | 50000 | 10000 |
| DTD | | 47 | 3760 | 1880 |
| ImageNet | | 1000 | 1281166 | 50000 |
| EuroSAT | | 10 | 18900 | 8100 |
| Resisc45 | Specialized | 45 | 6300 | 25200 |
| Pattern | | 38 | 24320 | 6080 |
| UCF | | 101 | 9537 | 3783 |

**Table 7: Comparison of computational complexity during the inference phase on 4 natural datasets. The results are conducted on an NVIDIA RTX 3090, wherein the B-B module is CLIP ViT-B/16, VPT denotes visual prompt tuning, and LORE indicates our Black-White Box model.**

| Method | Training Param. (%) | GFLOPs | FPS | Avg. acc. |
|--------|---------------------|--------|-----|-----------|
| B-B | - | 45.2 | 106.1 | 65.79 |
| B-B with VPT | 0.2 | 47.0 | 95.5 | 73.91 |
| LORE | 5.7 | 48.9 | 74.4 | 78.19 |