# OpenReview forum: "Enhancing Pre-trained ViTs for Downstream Task Adaptation: A Locality-Aware Prompt Learning Method"
_acmmm.org/ACMMM/2024/Conference — MM2024 Poster_

### Official Review · Reviewer_8sJP · 2024-05-21

**Rating:** 5
**Confidence:** 3

**Summary:**

The manuscript introduces a method called LORE, which leverages a knowledge-driven white box module to enhance local information for Vision Transformers (ViTs). In particular, the white box module introduces Knowledge-Locality Attention (KLA) to improve the local connections between image patches, utilizes Knowledge-Locality Attention (KLA) to identify important features based on the prompt, and incorporates a Prompt Generator (PG) to create the prompt, which is guided by knowledge locality matching. Comprehensive experiments illustrate the effectiveness of the LORE.

**Strengths:**

- Novelty and technical correctness: The method addresses the issue of insufficient local information in ViTs by enhancing local relationships and generating prompts explicitly.
- Comprehensive evaluation: The datasets and tasks used for evaluation are appropriate, and both quantitative and qualitative results confirm the effectiveness of the proposed method.
- Clarity: The writing and logic are clear, with the authors providing explanations for the motivation behind each module.
- Practical applications: The method is valuable for the implementation of ViTs in various downstream tasks.

**Limitations:**

1. Figure 2 is a bit confusing, It's difficult to differentiate different modules. The authors should consider adjusting the font size and color to enhance the clarity of the framework. Besides, the framework should show the training and inference process.
2. Table 5 lacks of ablation study on the Locality Interaction Network and Prompt Generator.

**Suitability:**

3

---

### Official Review · Reviewer_7C63 · 2024-05-22

**Rating:** 4
**Confidence:** 2

**Summary:**

This paper proposes a method, LOcality-aware pRompt lEarning (LORE), aiming to address the challenge of locality vanishing when adapting pre-trained ViTs to downstream tasks. LORE integrates a data-driven Black Box module (i.e.,a pre-trained ViT encoder) with a knowledge-driven White Box module which is a locality-aware prompt learning mechanism to compensate for ViTs’deficiency in incorporating local information. The results are good and comprehensive, which validates the efficacy of the proposed method.

**Strengths:**

1. The paper is well-written and easy to follow, the idea is intuitive and clear. Visualizations properly demonstrate the concept of the proposed framework.
2. The performances improvement of the proposed method are impressed.

**Limitations:**

1. It would be better to add domain generalization experiments to verify the robustness of the proposed method against distribution shifts, as demonstrated in ProGrad[1] and Maple[2].
2. It would be better to add cross-dataset transfer evaluation, as demonstrated in ProGrad[1] and Maple[2].
3. What impact do the hyper-parameter $\lambda$ and the dimension $r$ have on the performance of the proposed method? It would be better to provide a detailed explanation.
---
[1] Beier Zhu, Yulei Niu, Yucheng Han, Yue Wu, and Hanwang Zhang. 2023. Prompt-aligned Gradient for Prompt Tuning. In International Conference on Computer Vision.

[2] Muhammad Uzair Khattak, Hanoona Rasheed, Muhammad Maaz, Salman Khan, and Fahad Shahbaz Khan. 2023. Maple: Multi-modal prompt learning. In Proceedings of the IEEE/CVF Conference on Computer Vision and Pattern Recognition. 19113–19122.

**Suitability:**

3

---

### Official Review · Reviewer_EY6n · 2024-05-24

**Rating:** 5
**Confidence:** 3

**Summary:**

The paper addresses the limitation of Vision Transformers (ViTs) in extracting local information, which hampers their performance in downstream tasks. It introduces a LOcality-aware pRompt lEarning (LORE) method that integrates a data-driven Black Box (pre-trained ViT encoder) with a knowledge-driven White Box (locality-aware prompt learning mechanism). This includes a Locality Interaction Network (LIN) and Knowledge-Locality Attention (KLA) mechanism to enhance local relationships and capture critical local regions. By combining global and local information, LORE improves ViTs' performance across various downstream tasks.

**Strengths:**

- Overall the paper is a solid one. The method looks integral and makes sense.
- The visualization results demonstrate that the KLA mechanism effectively captures local information.
- The authors provide sufficient experiment and ablation studies to demonstrate the effectiveness of the proposed method.
- This paper is clear and well-written.

**Limitations:**

- Could you provide clarification on the process for generating negative prompts that demonstrate the Black Box module’s wrong understanding of images?
- Seems like the primary contribution comes from W-B. It is OK but somewhat limited in terms of novelty.
- Although K-L prototypes are not necessary for the inference phase, could you explain the extent to which the inference cost is increased by this method?
- Fig. 1 (c) might be better presented as a separate figure. Its connection to the rest of Figure 1 is not as strong as the connections among the other parts.

**Suitability:**

3

---

### Official Review · Reviewer_DTfR · 2024-05-24

**Rating:** 4
**Confidence:** 4

**Summary:**

This article introduces a test-time augmentation method named LORE, designed to optimize Vision Transformer (ViT) models. The LORE method effectively mitigates the locality vanishing problem commonly observed in ViT models when processing downstream datasets. The study conducts experimental evaluations on multiple downstream tasks and compares the results with various state-of-the-art (SOTA) methods, thereby demonstrating the effectiveness of the LORE method.

**Strengths:**

[+] The LORE method uniquely employs a combination of white-box and black-box approaches to alleviate the locality vanishing problem. The design principles of integrating a data-driven black box with a knowledge-driven white box are particularly intriguing.

[+] The authors conducted extensive experiments across various downstream tasks and datasets to demonstrate the effectiveness of their method. These tasks include few-shot image classification, easy-to-hard classification, image retrieval, point correspondences, and video object segmentation. The experimental results are quite compelling.

[+] The visualization of the model's attention effectively demonstrates the impact of LORE, supporting the authors' claimed innovations.

**Limitations:**

[-] The increase in parameter count and computational overhead introduced by the new module needs to be detailed. The authors should clarify the computational cost of the LORE module and analyze its impact on the model's inference speed.

[-] Attention should be paid to writing issues. For instance, references should be numbered according to their order of appearance. The presentation of Figure 2 is somewhat convoluted and might cause misunderstandings.

[-] The explanations of the equations are somewhat obscure. The equation descriptions in section 3.2.1 should include more detailed explanations, and the readability of the equations needs to be further improved to avoid any potential misunderstandings.

I'll raise my score if the authors can address my above concerns.

**Suitability:**

2

---

### Meta-Review · Area_Chair_Asac · 2024-06-29

**Recommendation:** Accept (Poster)
**Confidence:** 4

**Metareview:**

The paper initially got four consistent positive scores. The authors have provided a rebuttal. After checking the rebuttal, three of the reviewers are satisfied with it and all gave weak accept. However, the Reviewer 7C63 reduced the score from borderline accept to borderline reject due to lack of compared the generalization ability with related works. After carefully checked the paper and the rebuttal, the AC thinks this paper mainly focuses on the adaptation to downstream tasks and thus it is fine that missing the discussion about the generalization ability. The AC thus thinks this paper can be accepted as an effective task adaptation approach but strongly encourages the authors to provide the comparisons on generalization ability and cross-dataset evaluation, which can better reflect the effectiveness or drawback of the proposed method.